# Evaluating machine learning approaches for host prediction using H3 influenza genomic data

**Hoc Tran**[1]*, **Olaf Berke**[1], **Nicole Ricker**[2], **Zvonimir Poljak**[1]

**1** Department of Population Medicine, Ontario Veterinary College, University of Guelph, Ontario, Canada,
**2** Department of Pathobiology, Ontario Veterinary College, University of Guelph, Ontario, Canada

* htran10@uoguelph.ca

## Abstract

### Background

H3 influenza A viruses (IAV) have been shown to frequently cross the species barrier which can be an important factor in sustained transmission and spread. Machine learning methods have been widely explored for host prediction of IAV using genomic data; however, this is often done using data from only one of the eight IAV segments or by using all available IAV data to predict broad categories of hosts.

### Objective

The objective of this study was to combine machine learning algorithms with H3 IAV sequence data from all eight segments to train predictive machine learning models for distinct host prediction and validate model performance.

### Methods

Models were trained on both k-mers and amino acid properties alongside machine learning algorithms that included random forest and XGBoost for each of the eight IAV genome segments. Models were then validated on a test dataset through analytics of model class predicted probabilities and subsequently used to investigate between-species transmission patterns within case studies including canine H3N8, swine H3N2 2010.2, and duck H3 sequences.

### Results

Models demonstrated strong performance in host prediction across all eight segments on the test dataset, with overall accuracies and κ (kappa) values ranging from 0.995–0.997, 0.984–0.990, respectively. Misclassified test dataset sequences with high predicted probabilities (> 90%) were validated using available literature and were identified to be frequently associated with between-species transmission events. Between-species transmission patterns within case study model class

**Data availability statement:** All sequences obtained in this study are available from the GenBank database (accession numbers provided in supporting excel file.) at https://www.ncbi.nlm.nih.gov/genbank/.

**Funding:** Ontario Veterinary College PhD Scholarship - HT https://www.uoguelph.ca/ovc The funders had no role in study design, data collection and analysis, decision to publish, or preparation of the manuscript. Natural Sciences and Engineering Research Council of Canada (NSERC) Alliance - ZP Grant number: ALLRP 592576 - 23 https://www.nserc-crsng.gc.ca/Innovate-Innover/Alliance-Alliance_eng.asp The funders had no role in study design, data collection and analysis, decision to publish, or preparation of the manuscript.

**Competing interests:** The authors have declared that no competing interests exist.

predicted probabilities were also identified to be consistent with the literature in cases of both correct and incorrect classification.

## Conclusions

These models allow for rapid and accurate host prediction of H3 IAV datasets from any of the eight IAV segments and provide a solid framework that allows for identification of variants with higher than typical between-species transmission potential. However, results obtained on selected case studies suggest further improvements of the training and validation processes should be considered.

## Introduction

The influenza virus is a pathogen that is classified into the family *Orthomyxoviridae* and comprises four of its seven genera which each consist of one influenza virus species (A-D). Influenza A and B viruses are the primary agents responsible for causing seasonal epidemics in humans, with influenza A virus (IAV) being the only species to have caused pandemics to date [1,2]. Influenza C and D viruses are associated with more sporadic cases of infections, with influenza C viruses primarily affecting humans and influenza D viruses primarily affecting cattle [3,4]. Of these four species, IAV is generally considered the most important species in humans due to the global economic burdens associated with seasonal epidemics that result in over 650,000 deaths annually within human populations [5]. Similarly, significant economic losses are attributed to IAV infection in many different animal production industries that include poultry and swine, with millions of avians culled globally per annum; and an estimated cost of $1-$5 per market hog within the United States [6–8].

IAV can be classified into subtypes based on the antigenic properties of the outer surface glycoproteins of hemagglutinin (HA) and neuraminidase (NA), with 18 HA (H1-H18) and 11 NA (N1-N11) subtypes identified thus far, respectively [9]. H3 IAVs are of notable interest as they are often associated with more severe influenza seasons and also infect a very wide range of hosts that includes mammals such as humans, swine, canines, and equines in addition to avians such as chickens, mallards, and geese [1,10,11]. This wide host range has become an ongoing global concern to both public and animal health as many between-species transmission events that allow for novel strains to arise have been documented across the globe and these novel strains have been determined as the root cause of several past pandemics [10,12–15]. Spill-over events have been identified where H3 avian influenza viruses were found to be the origin of outbreaks within swine, canines, equines, and seals [16–19]. Several hundred cases of zoonotic infections in humans have also been identified to be caused by swine H3 IAVs in addition to numerous cases of swine IAV spilling over into turkeys [20–24]. Reverse zoonosis of human H3 IAVs has also been found to occur in swine with regularity [25–27]. Therefore, surveillance and timely identification of hosts with higher than typical potential for between-species transmission is important to limit the spread of H3 IAV and prevent the occurrence of future H3 IAV pandemics.

Large public repositories of whole genome sequence data from past IAV infections have become increasingly available over time as a result of advancements in next generation sequencing techniques, providing opportunities for machine learning techniques that require very large datasets. Machine learning algorithms such as eXtreme Gradient Boosting (XGBoost), random forest, and multinomial logistic regression with ridge penalization have previously been used in the classification of IAVs and prediction of their hosts/sources based on sequence data. However, this is often performed using methodology involving only the HA segment [28] or by using all subtypes with broad host categories such as avian [29–31]. Utilizing the full genome of IAV rather than only the HA segment has been demonstrated to result in more accurate surveillance and genetic characterization of IAV strains and allows for better understanding of reassortment of the remaining 7 segments which is often overlooked [32,33]. Furthermore, machine learning models have been previously trained on H3 HA sequence data from distinct avian classes such as turkeys, mallards, and chickens and predicted with high levels of accuracy, indicating that sequences from broad categories can be delineated into distinct species without degradation in host predictability [28]. Thus, the primary objective of this study was to combine machine learning algorithms with sequence data from all 8 segments of H3NX IAVs to train predictive machine learning models for distinct host prediction and validate model performance. Validated models were trained with the ultimate goal of contributing to the development of a framework for identifying IAV variants with higher than typical potential for between-species transmission.

## Materials and methods

### Dataset retrieval

The dataset used in this study was obtained on April 16th, 2024 by retrieving whole genome nucleotide and protein sequence sets from the National Center for Biotechnology Information (NCBI) Influenza Virus Database (IVD), Bacterial and Viral Bioinformatics Resource Center (BV-BRC) database, and Global Initiative on Sharing All Influenza Data (GISAID) Epiflu database [34–36]. Sequence selection criteria from each database are summarized in S1 File.

### Dataset preprocessing

Preprocessing of the dataset began by extracting the hosts from the labels of the 37328 sequence sets and assigning a host category as shown in Table 1. Whole genome sequence set duplicates within host species and across databases were checked for and excluded. Sequence sets from hosts where there were less than 100 sequences sets available were excluded as this was considered an insufficient amount of data for model training. Sequence sets from the avian category were excluded as avian is considered to be too broad of a category to be useful in host prediction. Sequence sets from the duck and goose categories were not excluded so that they could be later used as a case study and as an additional waterfowl class for model training, respectively (Table 1). After these exclusions, the dataset consisted of 7 host classes of canine, chicken, equine, goose, human, mallard, and swine for model training in addition to the 2 case study classes of duck and environment. Included species for each class are shown in S1 Table.

Sequence sets obtained for the combined IVD, BV-BRC, and Epiflu dataset initially, after preprocessing, after case study selection, and the final dataset used in model training/testing.

The dataset was then filtered by requiring each whole genome sequence set to have exactly one nucleotide sequence per genome segment (PB2, PB1, PA, HA, NP, NA, MP, NS) for a total of 8 nucleotide sequences. Protein sequences for the 10 essential proteins of PB2, PB1, PA, HA, NP, NA, M1, M2, NS1, NS2 and accessory protein PB1-F2 were also required for a total of 11 protein sequences. Protein sequences from the accessory protein PA-X were not included as preliminary analysis identified that PA-X protein sequences were absent in a significant portion of the genome sets (15%~).

Leading and lagging Ns within sequences were trimmed and whole genome sequence sets that contained a sequence with a N count of greater than 5% of the expected sequence length for any of the eight genome segments were excluded.

**Table 1. Species distribution of the dataset.**

| Species | Initial | Preprocessed | Case Study | Final Dataset for Training/Testing |
|---|---|---|---|---|
| Human | 29993 | 16566 | 0 | 16566 |
| Swine | 3272 | 2316 | 40 (H3N2 2010.2) | 2276 |
| Duck | 1324 | 977 | 977 (All) | 0 |
| Mallard | 838 | 713 | 0 | 713 |
| Equine | 637 | 168 | 0 | 168 |
| Canine | 410 | 294 | 33 (H3N8) | 261 |
| Chicken | 317 | 261 | 0 | 261 |
| Avian | 301 | 0 | 0 | 0 |
| Environment | 132 | 105 | 105 (All) | 0 |
| Goose | 37 | 29 | 0 | 29 |
| Turkey | 19 | 0 | 0 | 0 |
| Gull | 11 | 0 | 0 | 0 |
| Feline | 11 | 0 | 0 | 0 |
| Seal | 9 | 0 | 0 | 0 |
| Swan | 5 | 0 | 0 | 0 |
| Donkey | 3 | 0 | 0 | 0 |
| Unknown | 2 | 0 | 0 | 0 |
| Mink | 2 | 0 | 0 | 0 |
| Pheasant | 1 | 0 | 0 | 0 |
| Pelican | 1 | 0 | 0 | 0 |
| Ferret | 1 | 0 | 0 | 0 |
| Camel | 1 | 0 | 0 | 0 |
| Ostrich | 1 | 0 | 0 | 0 |
| Total | 37328 | 21429 | 1155 | 20274 |

Sequence sets that were identified to be lab strains or contained sequences that were shorter than 90% of the expected sequence length were excluded. Subtypes were also checked, and any sequence sets not belonging to the H3 subtype were excluded with the subtypes for the remaining 21429 sequence sets shown in S2 Table.

### Case study datasets

Four separate case study groups were extracted from the preprocessed dataset for model validation and investigation of between-species transmission patterns using model class predicted probabilities. The four case studies consist of sequence sets from 1) canine H3N8, 2) the swine H3N2 2010.2 clade, 3) duck H3NX, and 4) environment H3NX. Case study 1 was used to investigate canine H3N8, which was first detected in canines within the United States in 2004 and was identified to have spilled-over from equines into canines [37–39]. Case study 2 was used to investigate the swine H3N2 2010.2 clade, which was first detected in 2017 as the second occurrence of a distinct seasonal human H3N2 being transmitted from humans to swine within the United States [25,40]. Sequence sets for case study 2 were extracted by using the OctoFlu tool [41] to label all present H3 swine sequence sets and extracting only sets labelled as part of the H3N2 2010.2 clade. Case study 3 was used to investigate class predicted probabilities for the broad category of duck which contained sequence sets from several different species of ducks. Case study 4 was used to investigate the possible hosts for environmental H3NX sequence sets which were comprised of sequences not labelled with an animal host but with locations such as "environment" or "water" instead. Each case study dataset was separated by genome segment, with one dataset per genome segment.

### Training and test datasets

The remaining preprocessed dataset was randomly partitioned into a training dataset (70%) and a test dataset (30%). The training and test datasets were then separated by genome segment, with one training and test dataset per genome segment.

### Feature extraction

K-mers were extracted from both nucleotide and protein sequence using the Biostrings [42] and kmer [43] packages in R. K-mers of length 1–6 (n = 4, 16, 64, 256, 1024, 4096) were extracted from the nucleotide sequences and k-mers of length 1–3 (n = 20, 400, 8000) were extracted from the protein sequences. Amino acid properties (n = 6) were also extracted from the protein sequences using the alakazam package in R [44]. The chosen amino acid properties for feature extraction were amino acid length, gravy, bulk, aliphatic index, polarity, and overall net charge. Sequences from genome segments with one protein available (HA, NA, NP, PA, PB2) had a total of 13887 features extracted, whereas sequences from genome segments with two proteins available (PB1, NS, MP) had a total of 22314 features extracted. For models with two proteins available, protein 1 refers to PB1, NS1, MP1 and protein 2 refers to PB1-F2, NS2, and MP2.

### Feature selection

Feature selection was performed to reduce computational time in R and to use only the most important features from each genome segment for host prediction. This was done by training a preliminary random forest model per genome segment on all available features using the caret package in R with "rf" as the chosen algorithm and default parameters [45]. Cross-validation was not performed during feature selection to not further increase computational time. After the preliminary models were trained, feature importance was determined using mean decrease Gini scores [46] and features with the top 10% highest mean decrease Gini scores were selected to train the final models for each genome segment.

### Model training

Three models were trained for each of the eight genome segments for a total of 24 models. The three models per genome segment were trained using multinomial logistic regression with ridge penalization, random forest, and XGBoost which were implemented through the caret and glmnet packages in R [45,47,48]. Models were also cross-validated using 5-fold stratified cross-validation to ensure that all classes were represented proportionally due to the imbalanced nature of the dataset.

### Model hyperparameter tuning

Model hyperparameters were tuned using a grid search during cross-validation to obtain the most robust models possible. The $\lambda$ (regularization parameter) and $m_{try}$ (number of features randomly sampled at each split) hyperparameters were tuned for the multinomial logistic regression with ridge penalization and random forest models, respectively. For the XGBoost models, the hyperparameters consisting of the number of rounds (nrounds), learning rate (eta), maximum depth (max_depth), column subsampling (colsample_bytree), minimum sum of instance weight required within a child (min_child_weight), subsample ratio (subsample), and gamma (minimum split loss) were tuned in stepwise groups.

### Model testing

Trained models were validated on the test dataset to observe their performance on unseen data. Confusion matrices were generated during model validation and metrics that include overall accuracies, 95% confidence intervals, $\kappa$, no-information rates, p-values, and class sensitivities and specificities were calculated. $\kappa$ refers to Cohen's $\kappa$ which is calculated using the following equation:

$$\kappa = \frac{p_0 - p_e}{1 - p_e}$$

where $p_0$ is the observed agreement and $p_e$ is the expected agreement of the model [49]. No-information rate refers to the overall accuracy of a naïve classifier that predicts every input as the majority class.

The best performing model for each genome segment was then determined using overall accuracies and $\kappa$ values. In cases where two or more models had the same overall accuracies and $\kappa$ values within a genome segment, models using the algorithm with the best performance overall across all 8 segments were chosen as the best performing model. Host predictions from the best performing models for each segment were then tallied per whole genome sequence set (8 sequences, 1 per genome segment) and host prediction frequencies were investigated (e.g., all 8 segments correctly predicted as human for a human sequence set or with mixed host prediction, 7 segments correctly predicted as human, 1 segment predicted as swine).

## Model class predicted probabilities

Model class predicted probabilities were investigated by creating heatmaps with the predicted probabilities of correctly classified and misclassified sequences using only the best model per genome segment. Groups of sequences with available literature were labelled with the notation "Pattern #" on the heatmaps and were used for subsequent investigation of possible explanations for model predicted probabilities. Additionally, patterns in model class predicted probabilities were explored for each of the case study datasets by using the best performing models per genome segment on the case study datasets, and plotting the model class predicted probabilities averaged by year for case studies 1 and 2 and by sequence number for case studies 3 and 4.

Case study model predicted probabilities were further validated through phylogenetic analysis by constructing a representative maximum likelihood phylogenetic tree for the HA segment. Representative sequences were obtained for each of the 7 host classes and 4 case study datasets separately using the CD-HIT EST tool with a 90% similarity threshold [50]. Representative sequences were then aligned using MUSCLE with default settings and a maximum likelihood tree with 100 bootstrap iterations was constructed with default settings using MEGA11 software [51]. Patristic distances were then calculated using the ape package in R [52]. This tree was then imported into the Interactive Tree of Life (iTOL) tool for visualization [53].

## Results

### Model feature selection

The number of features pre- and post- feature selection is summarized in Table 2. Models with one protein (HA, NA, NP, PA, PB2) had 13887 features available and models with two proteins (PB1, NS, MP) had 22314 features available, which were reduced by 90% to 1388 and 2231 after performing feature selection, respectively. All models were found to select features from all nucleotide k-mer categories of size 1–6 and protein k-mer categories of size 1–3. Models with two proteins (PB1, NS, MP) were descriptively identified with a generally higher number of features selected in protein one (PB1, NS1, MP1) versus protein two (PB1-F2, NS2, MP2) with exceptions in the PB1 and MP models, where a higher number of amino acid properties was selected for the PB1-F2 protein versus the PB1 protein and a higher number of 1-mers was selected for the M2 protein versus the M1 protein, respectively.

### Amino acid properties

Amino acid properties selected by each model are summarized in Table 2. A minimum of 1 amino acid property was selected per protein for each of the models with the exception of the PB1 model where protein 1 had 0 amino acid properties selected. The HA, NA, and PB2 models differed from the other models by only having 1 amino acid property selected which was identified as net charge.

**Table 2. Number of features extracted from nucleotide and protein sequence sets and number of features selected after feature selection.**

| Feature | Sequence Type | Number of available features | Selected by HA model | Selected by NA model | Selected by NP model | Selected by PA model | Selected by PB2 model | Selected by PB1 model | Selected by NS model | Selected by MP model |
|---|---|---|---|---|---|---|---|---|---|---|
| 1-mer | Nucleotide | 4 | 3 | 2 | 4 | 4 | 4 | 2 | 4 | 4 |
| 2-mer | Nucleotide | 16 | 8 | 7 | 11 | 7 | 8 | 11 | 15 | 16 |
| 3-mer | Nucleotide | 64 | 32 | 29 | 37 | 31 | 30 | 46 | 55 | 61 |
| 4-mer | Nucleotide | 256 | 86 | 106 | 93 | 89 | 96 | 144 | 170 | 199 |
| 5-mer | Nucleotide | 1024 | 260 | 276 | 264 | 263 | 276 | 453 | 498 | 554 |
| 6-mer | Nucleotide | 4096 | 618 | 593 | 595 | 652 | 690 | 1149 | 964 | 997 |
| Amino acid length | Protein 1 | 1 | 0 | 0 | 0 | 0 | 0 | 0 | 1 | 0 |
| Amino acid gravy | Protein 1 | 1 | 0 | 0 | 1 | 1 | 0 | 0 | 1 | 1 |
| Amino acid bulk | Protein 1 | 1 | 0 | 0 | 1 | 1 | 0 | 0 | 1 | 1 |
| Amino acid aliphatic | Protein 1 | 1 | 0 | 0 | 1 | 1 | 0 | 0 | 1 | 1 |
| Amino acid polarity | Protein 1 | 1 | 0 | 0 | 0 | 1 | 0 | 0 | 1 | 1 |
| Amino acid net charge | Protein 1 | 1 | 1 | 1 | 1 | 1 | 1 | 0 | 1 | 1 |
| 1-mer | Protein 1 | 20 | 4 | 8 | 11 | 10 | 7 | 7 | 13 | 10 |
| 2-mer | Protein 1 | 400 | 107 | 90 | 106 | 96 | 77 | 82 | 103 | 77 |
| 3-mer | Protein 1 | 8000 | 269 | 276 | 263 | 231 | 199 | 194 | 259 | 142 |
| Amino acid length | Protein 2 | 1 | – | – | – | – | – | 0 | 0 | 0 |
| Amino acid gravy | Protein 2 | 1 | – | – | – | – | – | 0 | 1 | 1 |
| Amino acid bulk | Protein 2 | 1 | – | – | – | – | – | 0 | 0 | 1 |
| Amino acid aliphatic | Protein 2 | 1 | – | – | – | – | – | 0 | 0 | 1 |
| Amino acid polarity | Protein 2 | 1 | – | – | – | – | – | 1 | 1 | 1 |
| Amino acid charge | Protein 2 | 1 | – | – | – | – | – | 1 | 0 | 1 |
| 1-mer | Protein 2 | 20 | – | – | – | – | – | 6 | 9 | 14 |
| 2-mer | Protein 2 | 400 | – | – | – | – | – | 47 | 47 | 60 |
| 3-mer | Protein 2 | 8000 | – | – | – | – | – | 88 | 86 | 87 |
| Total | | 13887/22314[a] | 1388 | 1388 | 1388 | 1388 | 1388 | 2231 | 2231 | 2231 |

Feature selection was performed to keep the top 10% of features with the highest mean decrease Gini scores from preliminary random forest models trained on all available features from each genome segment.

[a] 13887 features if only one protein available, 22314 features if two proteins are available.

## Top ten most important features

The top 10 most important features selected by each of the preliminary models are summarized in S3 Table. A pattern was identified where models either had predominantly nucleotide 5-mers and 6-mers as the top 10 features (HA, NA, MP) or models had predominantly amino acid 2-mers and 3-mers as the top 10 features (NP, PA, PB2, NS). The PB1 model

differed by having a mix of both nucleotide and amino acid k-mers as the top 10 features. Additionally, the PB2 model was the only model with an amino acid property (net charge) within the top 10 features.

## Model validation results

Model performance metrics for all final trained models using the test dataset are summarized in Table 3. All models were shown to perform very well in host prediction across all 8 genome segments with overall accuracies and $\kappa$ values ranging from 0.9951–0.9967 and 0.9844–0.9896, respectively. Random forest and XGBoost models performed exactly the same in 5 segments (HA, NP, PB2, PB1, MP) where they had the same overall accuracies and $\kappa$ values. XGBoost models outperformed random forest models in two segments (NA, PA) and random forest model marginally outperformed XGBoost models in one segment (NS). Multinomial logistic regression with ridge penalization was found to be the worst performing model overall, performing worse than both XGBoost and random forest models across all genome segments except for the MP genome segment where all models were found to have the same overall accuracies and $\kappa$ values. All of the models were found to have significantly greater (p < 0.01) overall accuracies than no-information rates (0.8175). Macro F1 scores ranged from 0.8457–0.9240 for multinomial logistic regression models and 0.9251–9696 for XGBoost and random forest models. F1 scores for each class are summarized in S4 Table.

## Model class sensitivities

Model class sensitivities per genome segment are summarized in Table 4. Class sensitivities for the human, mallard, swine, and goose classes were shown to vary within each model per genome segment whereas the canine, chicken, and equine classes had little to no variation in model class sensitivities across all genome segments. Large variation was observed in class sensitivities for the goose class where class sensitivities ranged from 0.5000–0.7500 for the XGBoost models, 0.3750–0.7500 for the random forest models, and 0–0.3750 for the multinomial logistic regression with ridge penalization models. Model class specificities are summarized in S5 Table and ranged from 0.9874–1 across all classes, with slight variability observed in the human and mallard classes and little to no variability observed in the swine, goose, equine, chicken, and canine classes.

## Best performing models

The best performing models for the HA, NA, NP, PA, PB2, PB1, and MP genome segments were determined as the XGBoost models and the best performing model for the NS genome segment was identified as the random forest model. Host predictions from the best performing models per segment were then tallied for the each of the whole genome sequence sets within the test datasets and are shown in S6 Table. The majority of sequence sets were correctly classified with all segments predicted as the same host (6043/6078, 99.4%). A small number of sequence sets were classified as the correct host but with one or more segments predicted to be from a different host (13/6078, 0.213%), misclassified as the wrong host with all segments being predicted as the same host (15/6078, 0.247%), and misclassified as the wrong host with one or more segments being predicted as a different host (7/6078, 0.115%).

## Model class predicted probability heatmaps

For each genome segment a heatmap was generated to investigate model predicted probabilities of correctly classified and misclassified sequences. The heatmap for the HA genome segment is shown in Fig 1 with the heatmaps for the remaining genome segments shown in S1–S7 Figs. In Fig 1, the predicted probabilities for the correctly classified sequences are depicted in Fig 1A and the predicted probabilities for the misclassified sequences are depicted in Fig 1B. In Fig 1A, 6037/6055 (99.7%) sequences were classified with predicted probabilities of 90–100% for the correct class and 18/6055 (0.03%) were correctly classified with mixed host predicted probabilities. From the latter group of 18

**Table 3. Model performance metrics on the testing dataset.**

| Segment | Overall Accuracy | κ | 95% CI | No Information Rate (NIR) | P-Value [Acc > NIR] | Macro F1 |
|---|---|---|---|---|---|---|
| **HA Models** | | | | | | |
| RF | 0.9962 | 0.9880 | (0.9943, 0.9976) | 0.8175 | < 0.01 | 0.9252 |
| XGB | 0.9962 | 0.9880 | (0.9943, 0.9976) | 0.8175 | < 0.01 | 0.9355 |
| Ridge | 0.9951 | 0.9844 | (0.9930, 0.9967) | 0.8175 | < 0.01 | 0.8459 |
| **NA Models** | | | | | | |
| RF | 0.9964 | 0.9886 | (0.9945, 0.9977) | 0.8175 | < 0.01 | 0.9429 |
| XGB | 0.9965 | 0.9891 | (0.9947, 0.9979) | 0.8175 | < 0.01 | 0.9430 |
| Ridge | 0.9954 | 0.9854 | (0.9933, 0.9969) | 0.8175 | <0.01 | 0.8778 |
| **NP Models** | | | | | | |
| RF | 0.9957 | 0.9865 | (0.9937, 0.9972) | 0.8175 | < 0.01 | 0.9565 |
| XGB | 0.9957 | 0.9865 | (0.9937, 0.9972) | 0.8175 | < 0.01 | 0.9407 |
| Ridge | 0.9951 | 0.9844 | (0.9930, 0.9967) | 0.8175 | < 0.01 | 0.8776 |
| **PA Models** | | | | | | |
| RF | 0.9961 | 0.9875 | (0.9941, 0.9975) | 0.8175 | < 0.01 | 0.9689 |
| XGB | 0.9967 | 0.9896 | (0.9949, 0.9980) | 0.8175 | < 0.01 | 0.9696 |
| Ridge | 0.9952 | 0.9849 | (0.9932, 0.9968) | 0.8175 | < 0.01 | 0.8768 |
| **PB2 Models** | | | | | | |
| RF | 0.9962 | 0.9880 | (0.9943, 0.9976) | 0.8175 | < 0.01 | 0.9692 |
| XGB | 0.9962 | 0.9880 | (0.9943, 0.9976) | 0.8175 | < 0.01 | 0.9690 |
| Ridge | 0.9957 | 0.9865 | (0.9937, 0.9972) | 0.8175 | < 0.01 | 0.9240 |
| **PB1 Models** | | | | | | |
| RF | 0.9961 | 0.9875 | (0.9941, 0.9975) | 0.8175 | < 0.01 | 0.9251 |
| XGB | 0.9961 | 0.9875 | (0.9941, 0.9975) | 0.8175 | < 0.01 | 0.9351 |
| Ridge | 0.9956 | 0.9859 | (0.9935, 0.9971) | 0.8175 | < 0.01 | 0.8457 |
| **NS Models** | | | | | | |
| RF | 0.9962 | 0.9880 | (0.9943, 0.9976) | 0.8175 | < 0.01 | 0.9574 |
| XGB | 0.9961 | 0.9875 | (0.9941, 0.9975) | 0.8175 | < 0.01 | 0.9430 |
| Ridge | 0.9956 | 0.9860 | (0.9935, 0.9971) | 0.8175 | < 0.01 | 0.9036 |
| **MP Models** | | | | | | |
| RF | 0.9951 | 0.9844 | (0.9930, 0.9967) | 0.8175 | < 0.01 | 0.9330 |
| XGB | 0.9951 | 0.9844 | (0.9930, 0.9967) | 0.8175 | < 0.01 | 0.9338 |
| Ridge | 0.9951 | 0.9844 | (0.9930, 0.9967) | 0.8175 | < 0.01 | 0.8971 |

sequences, 7 sequences were identified to have mixed host predicted probabilities > 10% split between human and swine, 6 sequences split between goose and mallard, 2 sequences split between chicken, goose, human, and mallard, and 3 split between mallard and classes with less than 10% predicted probabilities. As an example, strain A/mallard/Hungary/19616/2007(H3N8) (accession # GQ240821) from Pattern 1 had predicted probabilities of 89% mallard, 7.2% goose, 2.1% human and 1.2% chicken. The remaining predicted probabilities for sequences from the mixed host Patterns 1–3 are detailed in S2 File.

In Fig 1B, 17/23 (73.9%) sequences were misclassified with predicted probabilities of 90–100% for the incorrect class and 6/23 (26.1%) were misclassified with mixed host predicted probabilities. From the former group of 17 sequences, 11 sequences were misclassified with 90–100% predicted probabilities as human, and 6 sequences were misclassified with 90–100% predicted probabilities as mallard. As an example, strain A/chicken/Nanjing/B854-2/2011(H3N8) (accession # KU158890) from Pattern 4 was being misclassified as mallard with 90–100% predicted probability and had predicted

**Table 4. Class sensitivities for each of the 7 classes on the testing dataset.**

| Segment | Canine | Chicken | Equine | Goose | Human | Mallard | Swine |
|---|---|---|---|---|---|---|---|
| **HA Models** | | | | | | | |
| RF | 1.000 | 0.9487 | 1.000 | 0.3750 | 0.9998 | 1.000 | 0.9809 |
| XGB | 1.000 | 0.9487 | 1.000 | 0.5000 | 0.9998 | 0.9953 | 0.9809 |
| Ridge | 1.000 | 0.9487 | 1.000 | 0 | 0.9998 | 1.000 | 0.9751 |
| **NA Models** | | | | | | | |
| RF | 1.000 | 0.9487 | 1.000 | 0.5000 | 0.9998 | 1.000 | 0.9809 |
| XGB | 1.000 | 0.9487 | 1.000 | 0.5000 | 0.9998 | 1.000 | 0.9824 |
| Ridge | 1.000 | 0.9487 | 1.000 | 0.1250 | 0.9998 | 0.9953 | 0.9780 |
| **NP Models** | | | | | | | |
| RF | 1.000 | 0.9487 | 1.000 | 0.6250 | 0.9992 | 0.9953 | 0.9795 |
| XGB | 1.000 | 0.9487 | 1.000 | 0.6250 | 0.9992 | 0.9906 | 0.9809 |
| Ridge | 1.000 | 0.9487 | 1.000 | 0.1250 | 0.9992 | 0.9953 | 0.9795 |
| **PA Models** | | | | | | | |
| RF | 0.9872 | 0.9487 | 1.000 | 0.7500 | 0.9998 | 0.9953 | 0.9780 |
| XGB | 0.9872 | 0.9487 | 1.000 | 0.7500 | 0.9998 | 1.000 | 0.9824 |
| Ridge | 0.9872 | 0.9487 | 1.000 | 0.1250 | 0.9994 | 1.000 | 0.9795 |
| **PB2 Models** | | | | | | | |
| RF | 0.9872 | 0.9487 | 1.000 | 0.7500 | 0.9998 | 0.9953 | 0.9795 |
| XGB | 0.9872 | 0.9487 | 1.000 | 0.7500 | 0.9998 | 0.9953 | 0.9795 |
| Ridge | 0.9872 | 0.9487 | 1.000 | 0.3750 | 0.9996 | 1.000 | 0.9795 |
| **PB1 Models** | | | | | | | |
| RF | 1.000 | 0.9487 | 1.000 | 0.3750 | 0.9998 | 1.000 | 0.9795 |
| XGB | 1.000 | 0.9487 | 1.000 | 0.5000 | 0.9998 | 0.9953 | 0.9795 |
| Ridge | 1.000 | 0.9487 | 1.000 | 0 | 0.9998 | 1.000 | 0.9795 |
| **NS Models** | | | | | | | |
| RF | 1.000 | 0.9487 | 1.000 | 0.6250 | 0.9996 | 1.000 | 0.9795 |
| XGB | 1.000 | 0.9487 | 1.000 | 0.5000 | 0.9998 | 0.9953 | 0.9795 |
| Ridge | 1.000 | 0.9487 | 1.000 | 0.2500 | 0.9994 | 1.000 | 0.9795 |
| **MP Models** | | | | | | | |
| RF | 0.9872 | 0.9487 | 1.000 | 0.5000 | 0.9992 | 0.9859 | 0.9795 |
| XGB | 0.9872 | 0.9487 | 1.000 | 0.5000 | 0.9992 | 0.9859 | 0.9795 |
| Ridge | 0.9872 | 0.9487 | 1.000 | 0.2500 | 0.9992 | 0.9953 | 0.9795 |

probabilities of 91% mallard, 2.8% human, 1.6% goose, 1.5% chicken, 1.5% canine, and 1% swine. The remaining predicted probabilities for misclassified sequences from Patterns 4–7 are detailed in S2 File.

## Case study predicted probabilities

The best performing models per genome segment were used on the case study datasets and model predicted probabilities were plotted averaged by year for case study 1 (canine H3N8) and case study 2 (swine H3N2 2010.2 clade) as shown in Figs 2-3, respectively, and plotted by sequence number for case study 3 (duck H3NX) and case study 4 (environment H3NX) as shown in Figs 4-5, respectively. In Fig 2, the 33 canine H3N2 sequence sets in case study 1 from 2004–2016 were misclassified as the equine class with high averaged predicted probabilities across all genome segments.

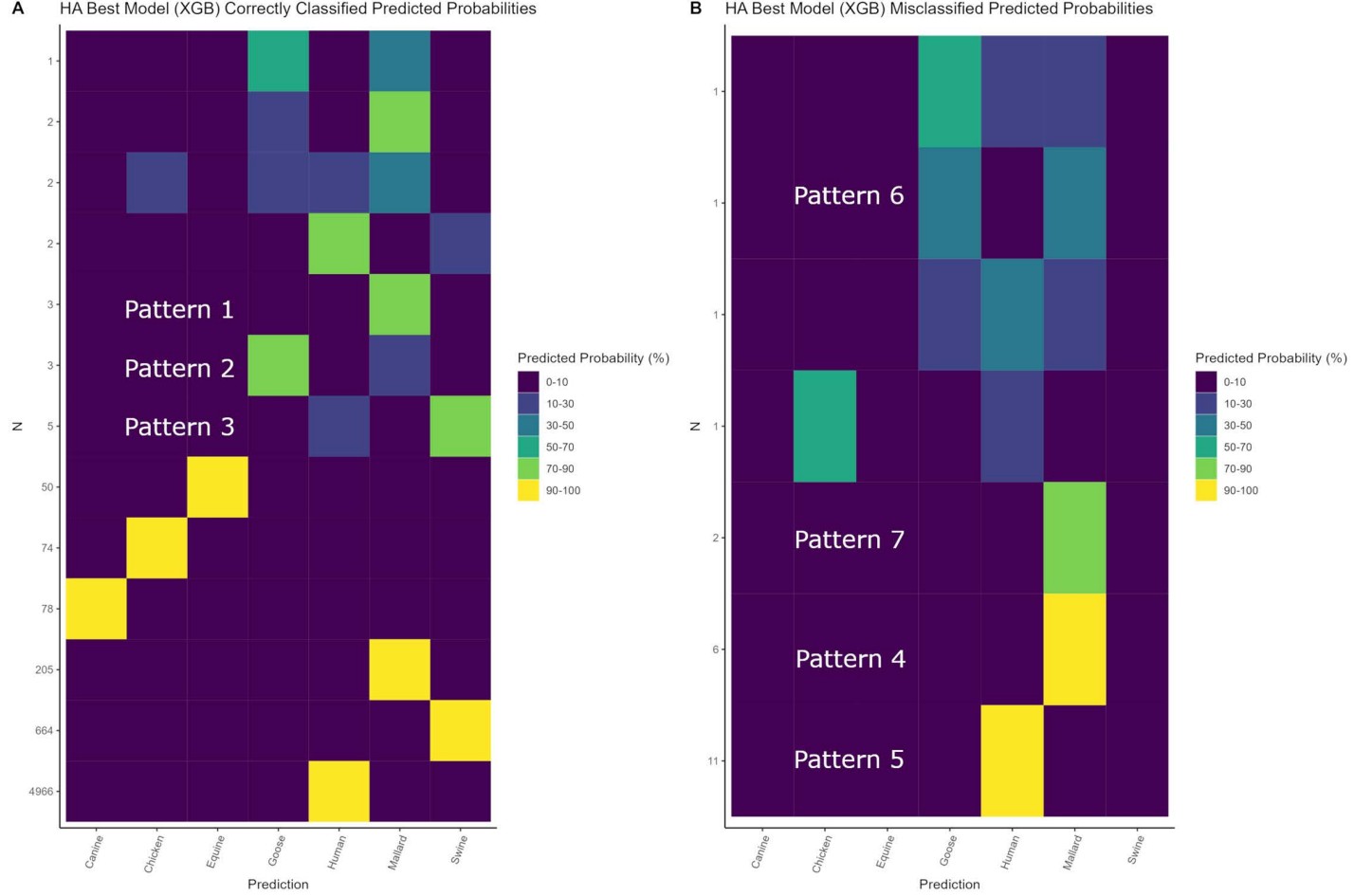

**Fig 1. Heatmaps of correctly classified and misclassified HA sequences.** Heatmaps for the HA genome segment using the XGBoost model (XGB), A shows the predicted probabilities for the correctly classified sequences and B shows the predicted probabilities for the misclassified sequences. Predicted probabilities are read as rows, and the N on the y axis denotes the number of sequences with the predicted probability pattern shown for the respective row. Rows labelled as Pattern 1-7 indicate rows where a representative sequence was investigated using available literature.

In Fig 3, the 40 H3N2 2010.2 clade sequence sets in case study 2 from 2017–2023 were misclassified as the human class with very high averaged predicted probabilities for the HA and NA segments with the remaining 6 segments being predicted as the swine class with very high averaged predicted probabilities. The HA and NA segments had predicted probabilities of 0.98–1 for the human class that stayed consistently the same throughout 2017–2023. The NP, PA, PB2, PB1, NS and MP segments had predicted probabilities of 0.75–0.80 for the swine class in 2017, which increased to 0.90–1 from 2018–2023.

In Fig 4, the 977 H3 duck sequence sets in case study 3 were predicted as the mallard class with high predicted probabilities for the majority of sequences across all genome segments. Predicted probabilities for the mallard class ranged from 0.75−1 for most sequence sets, with a moderate number of sequence sets predicted as the mallard class with predicted probabilities ranging from 0.25–0.75 for all genome segments. Smaller groups of sequences were also predicted as either the goose or chicken class with predicted probabilities ranging from 0.25–0.95 in each genome segment. Additionally, a few sequences were predicted as swine or human with predicted probabilities ranging from 0.25–0.85 in

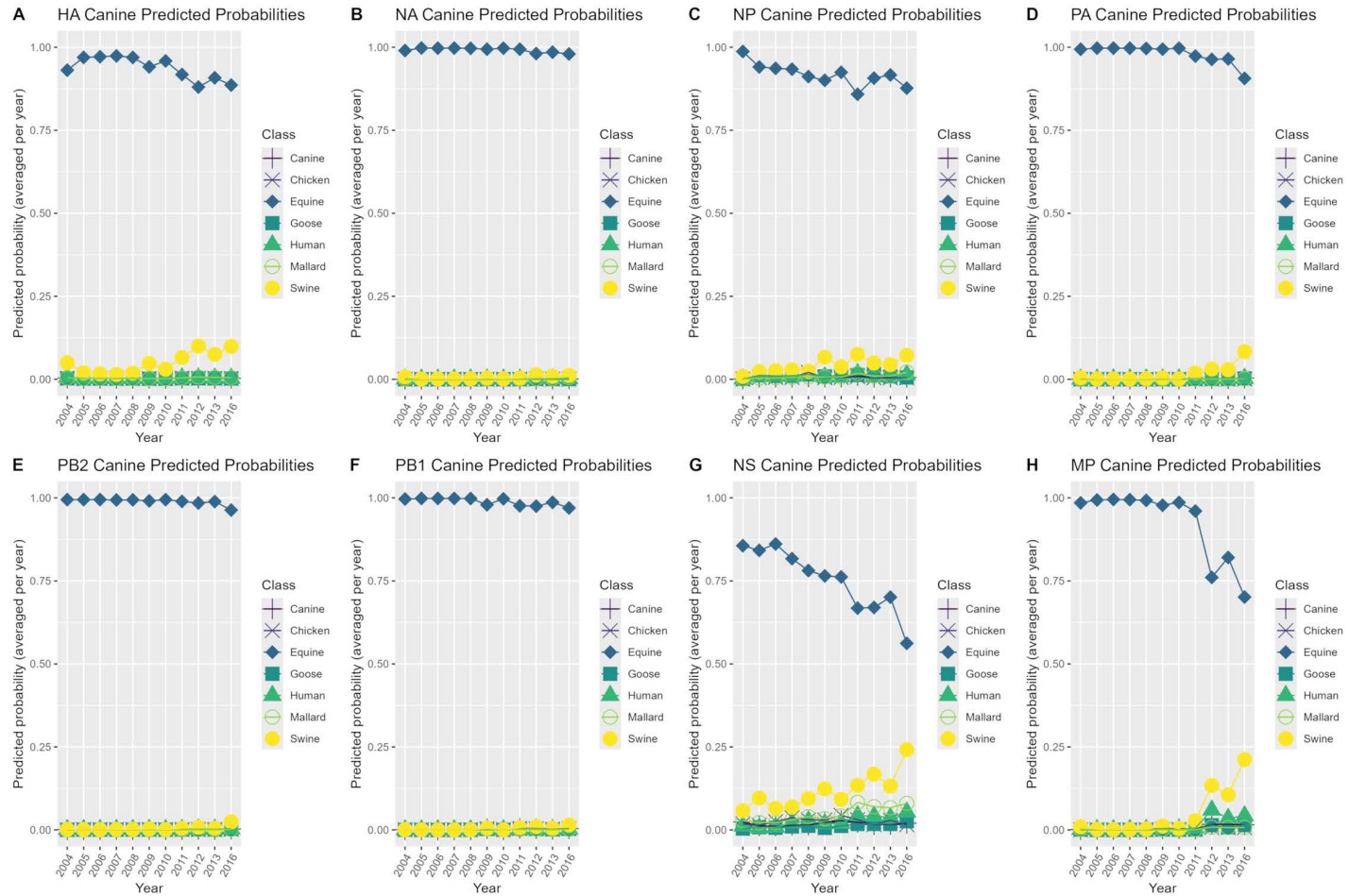

**Fig 2. Predicted probability plots by year for the canine H3N8 case study.** Averaged predicted probability plots by year for each genome segment using the best performing model per segment (random forest model for the NS segment, XGBoost models for the remaining 7 segments) on the canine H3N8 case study 1 dataset.

each genome segment, with a very small number also being predicted as canine with predicted probabilities ranging from 0.40–0.75 in the PA and PB1 segments.

In Fig 5, the 105 environment sequence sets in case study 4 were predicted as the mallard class with high predicted probabilities for the majority of sequence sets across all genome segments in addition to a small number of sequence sets being predicted as swine with high predicted probabilities. Predicted probabilities for the mallard class ranged from 0.5−1 for most sequence sets with a small group of sequence sets ranging from 0.7−1 for the swine class. The PB2 segment differed from the other segments by having a small number of sequences being predicted as the human class with predicted probabilities ranging from 0.27–0.80 and are shown in S7 Table.

## Phylogenetic analysis of case study datasets

Phylogenetic analysis was conducted using a maximum likelihood tree constructed from 64 representative HA sequences as shown in Fig 6. The representative H3N8 canine case study sequence was descriptively clustered with an equine representative sequence with accession number MH796298. The representative H3N2 2010.2 swine clade sequence was

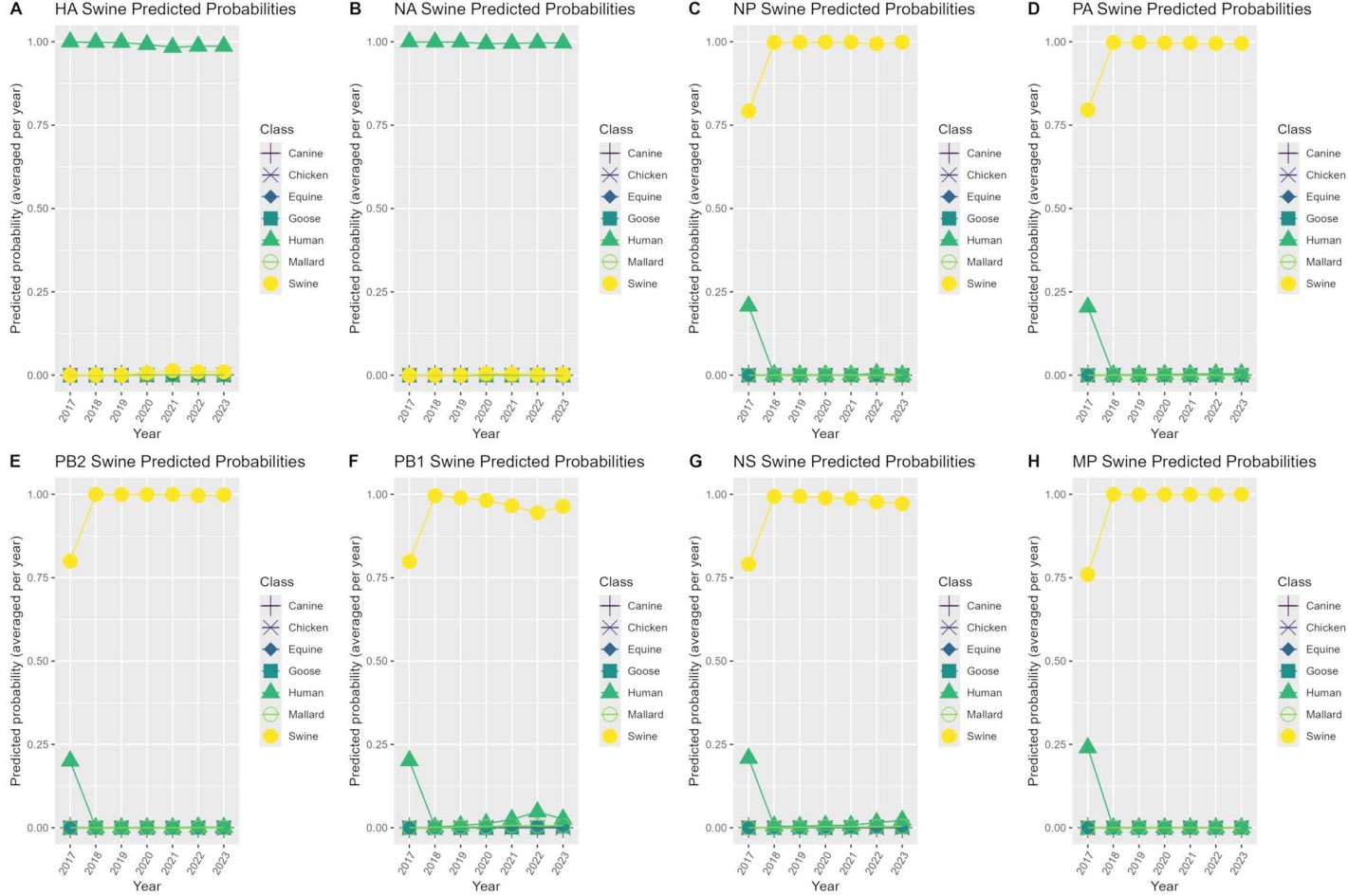

**Fig 3. Predicted probability plots by year for the swine H3N2 2010.2 case study.** Averaged predicted probability plots by year for each genome segment using the best performing model per segment (random forest model for the NS segment, XGBoost models for the remaining 7 segments) on the swine H3N2 2010.2 clade case study 2 dataset.

descriptively clustered with a human representative sequence with accession number KY925323. The duck case study had multiple representative sequences which descriptively formed many different clusters generally comprised of a mix of mallard, chicken, and goose representative sequences. The environment case study also had multiple representative sequences which descriptively formed clusters which generally consisted of mallard and duck representative sequences in addition to a few clusters with swine, human, or goose representative sequences. Summary statistics for patristic distances are shown in S8 Table. Species-species patristic distances are summarized in S9 Table, with patristic distances between and within each host class shown in S8 Fig.

## Discussion

Comparison of model performance during model validation revealed that all three types of models were capable of predicting the host for each genome segment with high accuracy using nucleotide and protein features. Random forest and XGBoost models had the same predictive performance in most genome segments, with XGBoost models slightly outperforming random forest in two segments whereas random forest outperformed XGBoost in only 1 segment. Prior studies

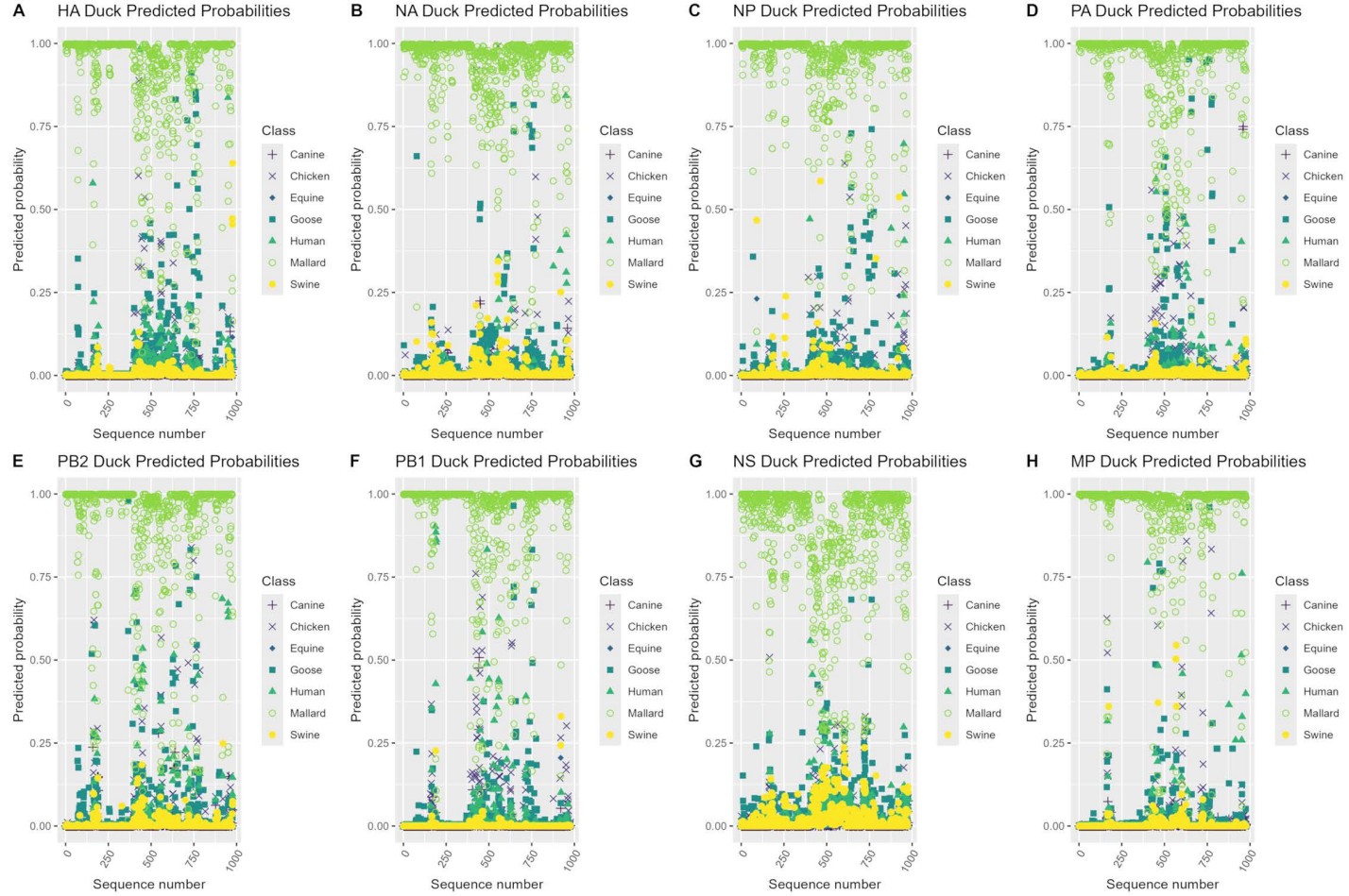

**Fig 4. Predicted probability plots by sequence number for the duck H3NX case study.** Predicted probability plots by sequence number for each genome segment using the best performing model per segment (random forest model for the NS segment, XGBoost models for the remaining 7 segments) on the duck H3NX case study 3 dataset.

have compared predictive capabilities of these models where similar results were found with boosting machines generally outperforming random forest models [28,54–56]. Notably, datasets from these other studies were often not imbalanced, whereas the models in our study were trained on an imbalanced dataset. Class sensitivities for our models were very high across all genome segments with the exception of the goose class, which had class sensitivities that were as low as zero for some of the multinomial logistic regression with ridge penalization models. This can be explained by the fact that the dataset was imbalanced and that there were only 29 sequence sets available for the goose class; however, sensitivities for goose ranged from 0.3750–0.7500 for the random forest and XGBoost models indicating that these ensemble methods may perform satisfactorily even when trained on a low number of sequence sets. Our results therefore support previous studies indicating that ensemble methods such as boosting machines and random forests are resilient options when working with imbalanced datasets [57–59].

Prior studies involving HA IAV classification have also achieved similar performance using different approaches with overall accuracies > 0.98 or F1 scores > 0.96 for predicting the host classes of human, swine, and the broad class of avian [29–31]. Usage of a broad avian class allows for a reduction in the number of classes to learn from and mitigates issues

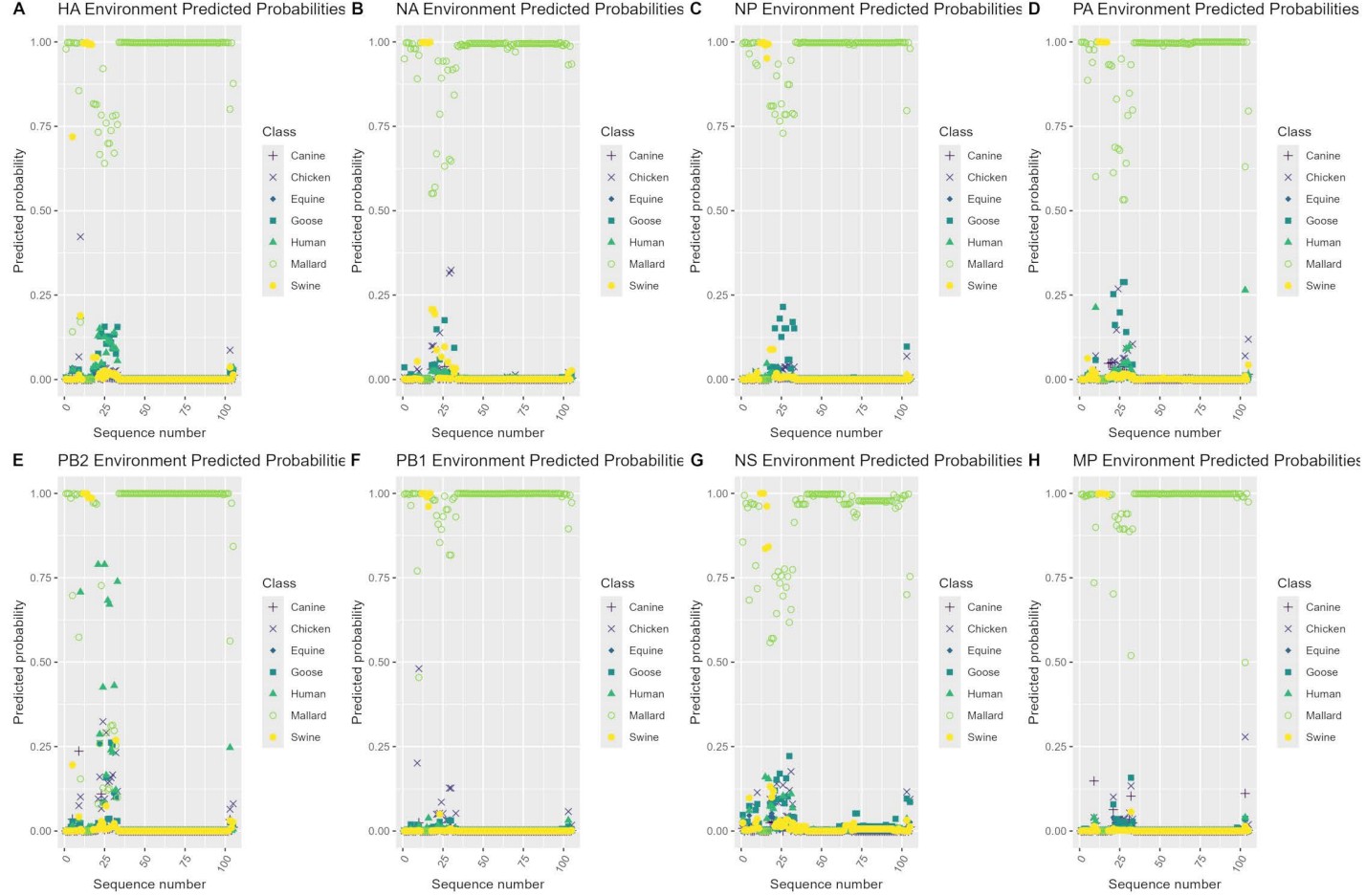

**Fig 5. Predicted probability plots by sequence number for the environment H3NX case study.** Predicted probability plots by sequence number for each genome segment using the best performing model per segment (random forest model for the NS segment, XGBoost models for the remaining 7 segments) on the environment H3NX case study 4 dataset.

associated with small sample size. Despite these advantages, this approach also results in the loss of species-level information and limits the ability to investigate between-species transmission patterns for many important avian hosts such as turkeys [21,60]. A recent study has demonstrated that H3 HA IAV avian sequences can be delineated into distinct species and predicted with moderate to high class sensitivities such as 0.84 for the chicken class and 0.57 for the turkey class [28]. The dataset in their study was imbalanced, with very few turkey sequences available (n = 26), and the low class sensitivity for the turkey class was thought to be due to small sample size. This result is very similar to the goose class in our present study, where the small sample size (n = 29) also resulted in low class sensitivity for the goose class across all of our models. Therefore, these results are indicative of the need for a large sample size when separating broad avian categories into distinct species for host prediction.

Many studies have used nucleotide sequences and nucleotide k-mers for host prediction of viruses, however, protein sequences and amino acid k-mers are less frequently used in comparison with even fewer studies using both in conjunction for host prediction [28,61–63]. Feature selection for our models demonstrated that k-mers from both nucleotide and protein sequences are useful for IAV genome segment host prediction, with all preliminary models selecting features from

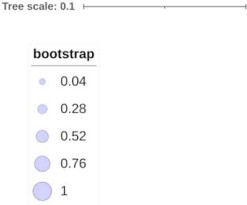

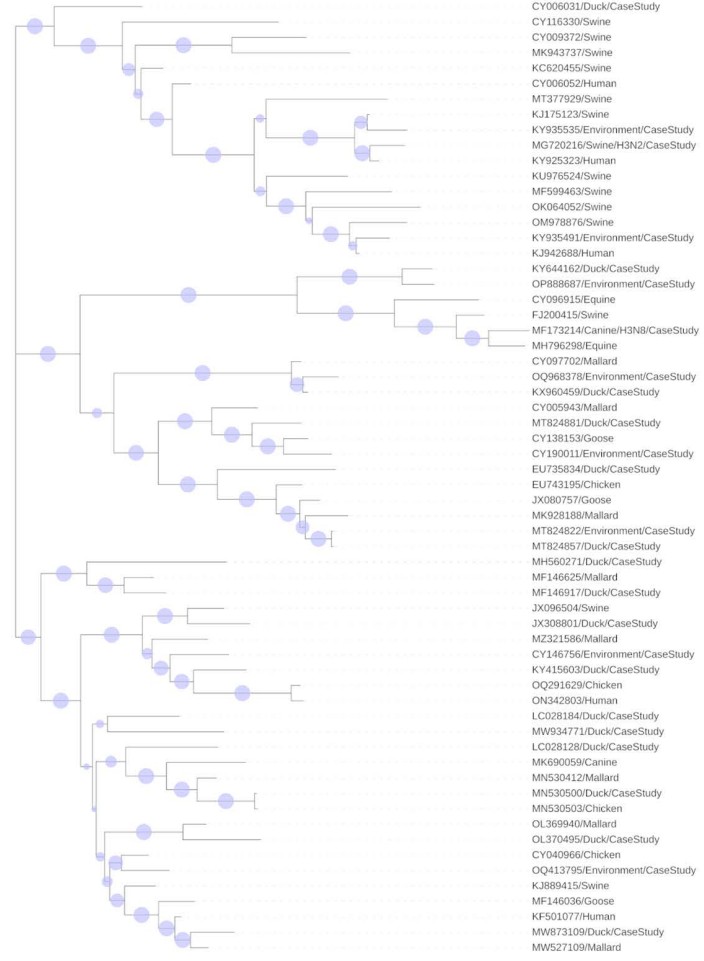

**Fig 6. Representative maximum-likelihood phylogenetic tree for HA sequences.** Maximum-likelihood phylogenetic tree with 100 bootstrap iterations constructed using the 64 HA representative sequences retrieved from the 4 case study datasets and 7 classes within the whole dataset excluding the case study sequences.

all available nucleotide and protein sequence k-mer categories. Protein sequences are more conserved than their nucleotide counterparts therefore features such as amino acid k-mers and amino acid properties may be more informative over wider evolutionary distances in host prediction [64]. Nucleotide k-mers of length 5–6 and amino acid k-mers of length 2–3 comprised the top 10 most important features in our models, with some models predominantly favouring nucleotide or amino acid k-mers. Optimal k-mer length has been shown to vary depending on the type of virus involved, with lengths of 2–4 shown to be optimal for a mix of RNA viruses in multi-host prediction and length >= 6 optimal for phages [63,65,66]. Longer k-mers may improve predictive performance as there could be host-specific k-mers present that can distinguish between hosts [67], which may explain why the longer k-mers were being selected as the top 10 most important features in our models. Of the six amino acid properties included, net charge was selected by 7 of the 8 preliminary models for protein 1, was the only amino acid property selected by the HA, NA, and PB2 models, and was the 2nd most important feature in the PB2 model. Net charge has been previously identified to alter the behaviour of electrostatic interactions involved in receptor and protein binding affinities for many genome segments including HA and PB2 in addition to playing a role in antigenic evolution of NA [68–71]. Previous studies have also identified net charge as a feature with high importance in

host prediction of the HA segment and in prediction of host tropism for the HA, NS1, and PB2 proteins [28,68]. Resultantly, amino acid net charge and feature importance in general are important characteristics that warrant further investigation in future studies involving host prediction of IAV genome segments.

Possible biological explanations were sought out within the literature to explain the predicted probabilities of correctly classified sequences with mixed host predictions and misclassified sequences from the HA segment. Sequences from Pattern 1–3 corresponded to correctly classified sequences and Pattern 4–7 corresponded to misclassified sequences from the HA heatmap (Fig 1). Sequence GQ240821 from Pattern 1 (89% mallard, 7.2% goose) was phylogenetically analysed and was found to cluster with various wild avian H3N8 isolates from Northern to Southern Europe [72]. Sequence AB569511 from Pattern 2 (85.3% goose and 13% mallard) clustered with waterfowl strains [73]. Sequence CY116315 from Pattern 3 (82.2% swine and 14.8% human) clustered with swine H3N2 human seasonal-origin strains [74]. Sequence KU158890 from Pattern 4 (91% mallard, sampled host was chicken) clustered with duck strains sampled within the same study and it was noted that the slaughterhouses where these samples were retrieved contained many different species of avians including chickens and ducks that remained in contact with each other for time periods of many days, providing a fertile environment for between-species transmission of IAV [75]. Sequence LC644998 from Pattern 5 (100% human, sampled host was swine) was found to cluster closely with seasonal human H3N2 strains, specifically seasonal human strains from within the same region (both pig and human strains were from Zambia) suggesting that reverse zoonosis had occurred [76]. Sequence JX080759 from Pattern 6 (47.4% mallard, 37.5% goose, sampled host was goose) was found to have been sampled from the Yukon-Kushokwim Delta and this region is considered a high priority for surveillance of between-species transmission due to it being a major migratory flyway across the eastern and western hemispheres [77]. Sequence JX096504 from Pattern 7 (81.6% mallard, 6.6% chicken, sampled host was swine) clustered with strains from domestic aquatic birds and was the first detection of a swine H3N2 strain with a genome segment from H5N1 highly pathogenic avian influenza [78]. As demonstrated, strong amounts of evidence from the literature support the model predicted probabilities for the HA segment and denote a pattern pertinent to risk assessment where misclassified sequences with high predicted probabilities are very often involved in between-species transmission. Furthermore, correctly classified and misclassified sequences with predicted probabilities that were very close between two hosts (e.g., Pattern 6, 47.4% mallard, 37.5% goose) represent sequences that warrant further investigation as these sequences may be suggestive of recent between-species transmission or potential for such further events.

Predicted probability plots for case study 1 revealed that all segments of canine H3N8 were being misclassified as the equine class with high averaged predicted probabilities from 2004–2016. Phylogenetic analysis using our maximum-likelihood tree supported these model predictions as the representative canine H3N8 HA sequence was found to cluster with a representative equine H3N8 HA sequence. Equine H3N8 was first isolated in equines in 1963 and was identified to have spilled-over into canines in 1999, with the first detection occurring in the United States in 2004 [37,39]. This subtype was primarily restricted to only the United States and seemingly became extinct from 2016 onwards, hence the limited time frame of 2004–2016 in our dataset. Phylogenetic analysis of equine and canine H3N8 conducted by a previous study determined that these strains were distinct, indicating that the virus had evolved in canines after introduction [79]. This was somewhat shown in the predicted probabilities plots for this case study where the predicted probabilities for equine slightly decreased and the predicted probabilities for swine were slightly increasing from 2008–2016, suggesting that changes were occurring in this canine subtype over this timeframe. Further investigation should be conducted to determine why the models were predicting this strain as becoming more swine-like rather than canine-like. Additionally, this misclassification by the models is indicative of the fact that the models are not capable of accurate host prediction for subtypes that are not present within the training dataset. However, this type of misclassification where all segments are being misclassified with high predicted probability is still informative when combined with phylogenetic analysis and could be useful in systematic identification of variants of interest if this is shown to be a recurrent pattern.

Predicted probability plots for case study 2 identified that the HA and NA segments of the swine H3N2 2010.2 clade were being misclassified with very high averaged predicted probabilities as the human class with the remaining 6 segments being predicted as the swine class with very high averaged predicted probabilities from 2017–2023. Phylogenetic analysis using our maximum-likelihood tree supported these model predictions as the representative swine H3N2 2010.2 clade HA sequence was found to cluster with a representative human H3N2 HA sequence. This strain was first detected in swine from Oklahoma in 2017 and the HA and NA of this strain was identified as having 99% nucleotide identity with seasonal human strains from 2017, suggesting that this strain was a product of reverse zoonosis [40]. This study also identified the remaining segments of PB2, PB1, PA, NP, and NS to be most similar to that of triple reassortment swine-origin strains with the MP segment phylogenetically analyzed as most similar to the 2009 H1N1 pandemic swine strains. Therefore, this evidence supports the model predicted probabilities for this case study and this misclassification where one or more segments are being misclassified with high predicted probability could be useful in systematic identification of reassortants if this pattern occurs consistently.

Predicted probability plots for case study 3 showed that a majority of duck H3 strains were being predicted with moderate to high predicted probabilities as the mallard class in addition to minor groups of sequences also being predicted as goose or chicken with moderate to high predicted probabilities across all genome segments. Phylogenetic analysis using our maximum-likelihood tree somewhat supported these model predictions as the representative duck HA sequences were descriptively clustered with a mix of mallard, goose, and chicken HA representative sequences. The duck case study was comprised of sequence sets from over 40 different species of ducks (S1 Table) excluding mallards, indicating that mallards may be representative of duck H3 strains as a whole. Mallards are the most common duck species found within the Northern Hemisphere and are key natural reservoirs for IAV infection [80,81] therefore it is logical that a majority of the duck sequence sets had high predicted probabilities for the mallard class. However, numerous amounts of sequence sets in this case study were simply labelled as "duck" therefore there is still some uncertainty whether these sequences sets are from mallards or other types of ducks. Furthermore, small groups of sequences being predicted as goose and chicken were not surprising given that these are the other two avian classes present in the models and are hosts that are commonly infected with avian IAV. Further investigations into the specific instances where sequence sets are being predicted as goose and chicken with high predicted probability may be useful to identify whether between-species transmission of strains had occurred in these instances.

Predicted probability plots for case study 4 are summarized in S3 File where a majority of the environmental sequences were being predicted as mallard, with a small group of sequence sets consistently predicted as swine across all segments. Most of the sequences predicted as mallard were identified as originating from Maryland, United States, with no further information available. On the other hand, six sequence sets were consistently predicted as swine across all 8 segments and were identified to have originated from Indiana, United States, with four of these sequence sets identified to be from agricultural fairs involved with swine [82] and two sequence sets identified to be from livestock exhibitions. A prior study had also predicted the same 4 sequence sets from agricultural fairs as swine using only the HA segment, supporting the model predictions for these environmental sequence sets as swine [28].

## Limitations

Case study 1 demonstrated a limitation of the models where no segments of canine H3N8 were being predicted as canine due to all canine H3N8 sequences being removed from the training dataset to be used as a case study. Evidently, these models require a minimum number of sequence sets from both a subtype and host to learn from before being able to accurately predict it, therefore new emerging subtypes and subtypes from classes outside of the ones included in model training will almost certainly be misclassified. However, misclassification of this type is useful in itself as these misclassified sequences can then be further investigated once flagged by these models as misclassified with high predicted probabilities.

Another limitation of these models would be that sequence data were insufficient or absent for many host classes of interest resulting in poor performance for the goose class or non-inclusion of important hosts such as turkeys. Class sensitivities for the goose class were much lower and inconsistent as a result of having only 29 sequence sets available for model training but would be expected to improve if the number of sequence sets obtained was at least 100. Turkeys have also been identified to be involved with H3 IAV between-species transmission events in many hosts including swine and wild waterfowl, warranting further investigation [21,60]. More data should therefore be collected from additional databases to retrieve as many sequence sets as possible to maximize predictive performance and to ensure that all important hosts are included as classes for model prediction. Alternative approaches to this may include applying oversampling methods such as synthetic minority oversampling technique (SMOTE) when databases are exhausted and additional sequence sets are not available [83–85]. Additionally, recombination of sequences was not considered within this study when it is an important feature involved in viral evolution and should correspondingly be investigated within future studies regarding host prediction [86].

Furthermore, an additional limitation would be that the features chosen during feature selection may be biased towards our models as cross-validation was not performed during feature selection. This results in lower generalizability of our features to other models. Future studies should therefore look to perform cross-validation while conducting feature selection to obtain the most robust and unbiased features possible for subsequent model training. Other approaches to cross-validation and model validation should also be considered to limit the possibility of data leakage when splitting the data randomly into training and test sets.

## Conclusion

In conclusion, all models demonstrated strong performance in distinct host prediction of IAV whole genome sequence data using features comprised solely of k-mers and amino acid properties with overall accuracies and κ values greater than 0.995 and 0.984, respectively. Models involving the ensemble methods of random forest and XGBoost were also shown to be resilient options for host prediction of very small minority classes in imbalanced datasets as these models had class sensitivities that ranged from 0.375–0.750 for the smallest minority class of goose. In comparison, models involving the non-ensemble method of multinomial logistic regression with ridge penalization had lower class sensitivities that ranged from 0–0.375. Furthermore, misclassified sequence sets with high predicted probabilities were identified as possible indicators for systematic identification of between-species transmission events and reassortant strains and were strongly supported by external validation using past literature and case study datasets. Similarly, correctly classified and misclassified sequences with predicted probabilities that were very close for two or more hosts were also indicative of recent and potential between-species transmission events. Application of these models as classifiers for H3 IAVs will therefore allow for accurate and rapid host prediction of IAV sequence data from any of the 8 genome segments and provide a strong framework that can be expanded upon for risk assessment and investigation of variants with higher than typical potential for between-species transmission. Nonetheless, results on specific case studies which resulted in misclassification also warrant caution, and further improvement of the training and validation process to prevent data leakage between the training and the validation datasets.

## Supporting information

**S1 Table. Sequence labels included from each class.** Sequence labels included from each species class after extracting the hosts from the labels and preprocessing to only have the classes of canine, equine, goose, human, mallard, swine, duck, and environment.
(DOCX)

**S2 Table. Subtype distribution after preprocessing.** Subtype distribution of the 21429 H3 whole genome sequence sets by host class retrieved from the NCBI Influenza Virus Database, BV-BRC database, and EpiFlu database after preprocessing was completed.
(DOCX)

**S3 Table. Top 10 most important features for each genome segment.** Top 10 features with the highest mean decrease Gini scores after feature selection for each of the 8 preliminary random forest models.
(DOCX)

**S4 Table. F1 Scores for each class on the testing dataset.** F1 scores for each of the 7 classes of canine, chicken, equine, goose, human, mallard, and swine obtained during model validation using the trained models from each genome segment on their respective test datasets.
(DOCX)

**S5 Table. Class specificities for each class on the testing dataset.** Class specificities for each of the 7 classes of canine, chicken, equine, goose, human, mallard, and swine obtained during model validation using the trained models from each genome segment on their respective test datasets.
(DOCX)

**S6 Table. Segment host prediction counts on the testing dataset.** Segment host prediction counts obtained from using the best performing models per genome segment (random forest model for the NS segment, XGBoost models for the remaining 7 segments) on the test dataset of 6078 sequences.
(DOCX)

**S7 Table. Accession numbers and labels for PB2 environment sequences being predicted as human.**
(DOCX)

**S8 Table. Summary statistics for patristic distances from the maximum-likelihood phylogenetic tree.** Mean, median, mode, minimum, and maximum patristic distances were calculated from the representative HA maximum-likelihood phylogenetic tree.
(DOCX)

**S9 Table. Summary statistics for species-species patristic distances.** Mean, median, and patristic distance range for each of the species-species pairs.
(DOCX)

**S1 Fig. Heatmaps of correctly classified and misclassified PB2 sequences.** Heatmaps for the PB2 genome segment using the XGBoost model (XGB), A shows the predicted probabilities for the correctly classified sequences and B shows the predicted probabilities for the misclassified sequences. Predicted probabilities are read as rows, and the N on the y axis denotes the number of sequences with the predicted probability pattern shown for the respective row.
(TIFF)

**S2 Fig. Heatmaps of correctly classified and misclassified PB1 sequences.** Heatmaps for the PB1 genome segment using the XGBoost model (XGB), A shows the predicted probabilities for the correctly classified sequences and B shows the predicted probabilities for the misclassified sequences. Predicted probabilities are read as rows, and the N on the y axis denotes the number of sequences with the predicted probability pattern shown for the respective row.
(TIFF)

**S3 Fig. Heatmaps of correctly classified and misclassified PA sequences.** Heatmaps for the PA genome segment using the XGBoost model (XGB), A shows the predicted probabilities for the correctly classified sequences and B shows the predicted probabilities for the misclassified sequences. Predicted probabilities are read as rows, and the N on the y axis denotes the number of sequences with the predicted probability pattern shown for the respective row.
(TIFF)

**S4 Fig. Heatmaps of correctly classified and misclassified NP sequences.** Heatmaps for the NP genome segment using the XGBoost model (XGB), A shows the predicted probabilities for the correctly classified sequences and B shows the predicted probabilities for the misclassified sequences. Predicted probabilities are read as rows, and the N on the y axis denotes the number of sequences with the predicted probability pattern shown for the respective row.
(TIFF)

**S5 Fig. Heatmaps of correctly classified and misclassified NA sequences.** Heatmaps for the NA genome segment using the XGBoost model (XGB), A shows the predicted probabilities for the correctly classified sequences and B shows the predicted probabilities for the misclassified sequences. Predicted probabilities are read as rows, and the N on the y axis denotes the number of sequences with the predicted probability pattern shown for the respective row.
(TIFF)

**S6 Fig. Heatmaps of correctly classified and misclassified MP sequences.** Heatmaps for the MP genome segment using the XGBoost model (XGB), A shows the predicted probabilities for the correctly classified sequences and B shows the predicted probabilities for the misclassified sequences. Predicted probabilities are read as rows, and the N on the y axis denotes the number of sequences with the predicted probability pattern shown for the respective row.
(TIFF)

**S7 Fig. Heatmaps of correctly classified and misclassified NS sequences.** Heatmaps for the NS genome segment using the random forest model (RF), A shows the predicted probabilities for the correctly classified sequences and B shows the predicted probabilities for the misclassified sequences. Predicted probabilities are read as rows, and the N on the y axis denotes the number of sequences with the predicted probability pattern shown for the respective row.
(TIFF)

**S8 Fig. Between and within species patristic distances for each of the 7 classes.** Patristic distances between and within each of the 7 host classes plotted separately.
(TIFF)

**S1 File. Supplementary information on the sequence selection criteria for each of the databases.**
(DOCX)

**S2 File. Supplementary information on the Fig 1A and 1B heatmap predicted probabilities.**
(DOCX)

**S3 File. Supplementary information on the environment case study.**
(DOCX)

## Acknowledgments

We would like to thank researchers for depositing their sequences into public repositories allowing for ease of access.

## Author contributions

**Conceptualization:** Zvonimir Poljak.

**Data curation:** Hoc Tran.

**Formal analysis:** Hoc Tran.

**Funding acquisition:** Zvonimir Poljak.

**Investigation:** Hoc Tran.

**Methodology:** Hoc Tran.

**Project administration:** Zvonimir Poljak.

**Software:** Hoc Tran.

**Supervision:** Olaf Berke, Nicole Ricker, Zvonimir Poljak.

**Validation:** Hoc Tran.

**Visualization:** Hoc Tran.

**Writing – original draft:** Hoc Tran.

**Writing – review & editing:** Olaf Berke, Nicole Ricker, Zvonimir Poljak.

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
