## [Decision Letter · Decision Letter 0]

8 Aug 2025

Dear Dr. Tran,

Thank you for submitting your manuscript to PLOS ONE. After careful consideration, we feel that it has merit but does not fully meet PLOS ONE’s publication criteria as it currently stands. Therefore, we invite you to submit a revised version of the manuscript that addresses the points raised during the review process.

We look forward to receiving your revised manuscript.

Kind regards,

Victor C Huber

Academic Editor

PLOS ONE

Journal Requirements:

Additional Editor Comments:

During the revision process, please pay attention to comments related to the overall presentation of the approach taken, the data collected, and the discussion of your findings.

Reviewers' comments:

Reviewer's Responses to Questions

**Comments to the Author**

1. Is the manuscript technically sound, and do the data support the conclusions?

Reviewer #1: Yes

Reviewer #2: Yes

2. Has the statistical analysis been performed appropriately and rigorously?

Reviewer #1: Yes

Reviewer #2: Yes

3. Have the authors made all data underlying the findings in their manuscript fully available?

Reviewer #1: Yes

Reviewer #2: Yes

4. Is the manuscript presented in an intelligible fashion and written in standard English?

Reviewer #1: Yes

Reviewer #2: Yes

Reviewer #1: The manuscript addresses an important and relevant question in the area of viral host prediction using ML models. The integration of k-mer and amino acid property features across all 8 genome segments is commendable. However, I have significant concerns about the technical rigor, overfitting risk, and lack of key methodological clarifications that must be addressed before publication.

1. Data leakage risk and overfitting

• The reported accuracy (>99%) and kappa values (>0.98) are suspiciously high for real-world genomic host prediction tasks, especially with unbalanced classes (e.g., only 29 goose sequences). It is unclear how potential data leakage between training and test sets was prevented, especially for highly similar or duplicate strains.

• Please clarify how the training/test split was done. Were related strains, or strains sampled at different time points from the same lineage, present in both splits?

2. Class imbalance treatment

• The manuscript mentions imbalanced data but lacks a robust strategy to address it. Did the authors use any resampling, class weights, or other techniques besides stratified folds?

• Given that some classes have <30 samples (e.g., goose), any claim about high sensitivity should be treated with caution. Consider using metrics robust to imbalance such as macro-F1 and AUC per class.

3. Statistical rigor

• The manuscript reports high kappa and accuracy but does not provide sufficient validation to prove generalizability. Did you run independent external validation (e.g., on isolates from a different region or time period)?

• No confidence intervals or variance estimates for feature importance or predicted probabilities are provided.

• The phylogenetic validation is interesting but is more descriptive than statistical; quantitative phylogenetic distance metrics would strengthen this.

4. Feature selection and reproducibility

• The feature space is huge (~22k features for some segments). Reducing to the top 10% of features based on random forest Gini scores is sensible but must be done entirely inside the CV loop to avoid bias. Please clarify this.

• The feature importance results need more biological interpretation: why are certain k-mers or AA properties dominant?

Reviewer #2: This manuscript presents a timely and well-organized study using machine learning to predict host species of H3 influenza A viruses based on full genome sequence data. The authors collected a large and diverse dataset, applied multiple machine learning models (including random forest and XGBoost), and explored both nucleotide and protein-based features to train their models. They also tested the models in several relevant case studies, including known cross-species transmission events, which adds practical value to their findings.

The key strength of this study is that it uses sequence data from all eight influenza genome segments, rather than focusing only on the HA gene. This broader approach is important and helps capture more of the viral genome’s evolutionary signals related to host adaptation. The addition of protein features, such as amino acid properties, particularly net charge, is another nice aspect that supports the biological relevance of the models.

However, I think the manuscript could benefit from more clarity on a few points related to novelty and broader significance. While the technical work is strong, it’s not always clear how this model improves upon existing host prediction tools or previously published methods. Including a direct comparison with other approaches would help establish the impact of the current work. Also, while the overall accuracy is very high, the performance is lower for smaller classes like goose, and this should be discussed more clearly, especially given the imbalanced dataset.

One other concern is the decision to exclude the avian group as a whole, while keeping goose and mallard as separate classes. Since avian hosts play a major role in influenza ecology, some explanation of this choice and its limitations would be helpful. Including a brief discussion of how this might affect generalizability would improve the manuscript.

Finally, while the results are comprehensive, the writing - particularly in the Methods and Results sections - is sometimes too dense and overly detailed. Condensing some technical sections and focusing more on the biological interpretation and practical use of the models (for example, in surveillance or risk assessment) would help improve readability and highlight the real-world relevance of the work.

In summary, this is a technically solid and timely study with high potential. With better framing of the novelty, clearer discussion of limitations, and some refinements to the writing, I believe this manuscript could make a valuable contribution to the field of influenza research and viral host prediction.

**Do you want your identity to be public for this peer review?** For information about this choice, including consent withdrawal, please see our Privacy Policy

Reviewer #1: No

Reviewer #2: No

---

## [Author Response · Author response to Decision Letter 1]

8 Oct 2025

We would like to thank the editor and reviewers for providing their comments for our manuscript submission. We have responded to all reviewer comments in the decision letter through the Response to the Reviewers file attached as a word document.

Reviewer #1: The manuscript addresses an important and relevant question in the area of viral host prediction using ML models. The integration of k-mer and amino acid property features across all 8 genome segments is commendable. However, I have significant concerns about the technical rigor, overfitting risk, and lack of key methodological clarifications that must be addressed before publication.

1. Data leakage risk and overfitting

• The reported accuracy (>99%) and kappa values (>0.98) are suspiciously high for real-world genomic host prediction tasks, especially with unbalanced classes (e.g., only 29 goose sequences). It is unclear how potential data leakage between training and test sets was prevented, especially for highly similar or duplicate strains.

We thank reviewer 1 for their comments. The reported overall accuracies and kappa values are indeed >0.98 for both metrics, however, we would disagree that they are suspiciously high for multiple reasons.

The no-information rate was 0.8175 (human class) therefore the baseline accuracy for a naïve model would be 81.75% if it were to simply predict everything as human. Given that the majority of the dataset is human, the overall accuracy is indeed highly biased towards the model performing very well on predicting the human class which is not surprising given the abundance of human sequence data available. The remaining 6 classes comprised the remaining 18.25% of the dataset, with each class having at least 100 sequences with the exception of goose which had 29 as mentioned.

We agree that high overall accuracies can therefore be misleading as a metric on its own on an imbalanced dataset, which is why we also looked at Kappa, class sensitivities, and class specificities to determine how well the models are performing across all classes and provided them alongside the overall accuracies. High kappa values are especially informative of model performance within imbalanced multi-class classifiers as it is not as biased towards the majority class by taking class distributions into consideration and measures expected agreement vs that of by chance, providing stronger evidence alongside overall accuracies of strong model performance across all classes.

Prior studies for IAV host prediction have also been able to achieve overall accuracies > 0.98 for multiclass IAV classification using only the HA genome segment (Chrysostomou et al. 2021, Alberts et al., 2024).

We address data leakage and how training/test data were split in the following comment.

Chrysostomou C, Alexandrou F, Nicolaou MA, Seker H. Classification of Influenza Hemagglutinin Protein Sequences using Convolutional Neural Networks. Proceedings of the Annual International Conference of the IEEE Engineering in Medicine and Biology Society, EMBS. 2021; 1682–1685. doi:10.1109/EMBC46164.2021.9630673

Alberts F, Berke O, Maboni G, Petukhova T, Poljak Z. Utilizing machine learning and hemagglutinin sequences to identify likely hosts of influenza H3Nx viruses. Prev Vet Med. 2024;233: 106351. doi:10.1016/J.PREVETMED.2024.106351

• Please clarify how the training/test split was done. Were related strains, or strains sampled at different time points from the same lineage, present in both splits?

Training and test datasets were formed by splitting the data using the basic method of randomly splitting the data into 70% training and 30% testing. It is therefore possible for related strains or strains sampled at different time points from the same lineage to be present in both training and test datasets.

We agree with the reviewer that by doing it in this manner, data leakage is a possibility where very similar sequences could end up in both the training and test datasets, resulting in overly optimistic performance of the trained model on the test dataset.

Resultantly, this is one of the primary reasons as to why we included the case studies comprised of H3N2 2010.2 clade in swine and H3N8 in canines for additional model validation as these case studies contain sequence data that is not present within the training datasets at all, therefore data leakage should not occur when validating model performance on these case studies. Evaluation of the performance of these models on these case studies was informative and indeed pointed to careful application of these models and need to further work on adequate training and validation.

Furthermore, another reason why the datasets were split in this manner was that identifying sequences into lineages or clades for each host class is not an easily done task. There are tools that can label human and swine sequences into clades and lineages (NextClade, OctoFlu), however, there are no tools available that can label them for the remaining 5 classes. While considerable work has been done on other subtypes that are frequently reported to be detected in multiple host species (and other categories), surprisingly it is not uniformly done for H3 viruses across multiple hosts.

A phylogenetically informed approach would be ideal when splitting the data into training and test datasets and is something that I am currently working on as part of my PhD thesis, where I am using clustering methods with various sequence similarity thresholds to construct a maximum-likelihood phylogenetic tree for all 7 classes to label them into clades/lineages. Development and application of this methodology will need more time before it can be applied to these models. We now mention that further improvements to model validation need to be considered on L39-43 in the abstract, L586-588 in the limitations, and L608-610 in the conclusion.

Aksamentov et al., (2021). Nextclade: clade assignment, mutation calling and quality control for viral genomes. Journal of Open Source Software, 6(67), 3773, https://doi.org/10.21105/joss.03773

Chang J, Anderson TK, Zeller MA, Gauger PC, Vincent AL. octoFLU: Automated Classification for the Evolutionary Origin of Influenza A Virus Gene Sequences Detected in U.S. Swine. Microbiol Resour Announc. 2019;8: e00673-19. doi:10.1128/MRA.00673-19

2. Class imbalance treatment

• The manuscript mentions imbalanced data but lacks a robust strategy to address it. Did the authors use any resampling, class weights, or other techniques besides stratified folds?

Stratified 5-fold cross validation was the sole method used to address the imbalanced dataset during model training as it ensured that the same proportion of each class would be present within each fold as some classes may be absent in some folds if standard cross-validation was performed.

We agree that there are various methods that could have been used to further address this such as down sampling the human class rather than including all of the human sequences for model training, however, we chose not to employ these strategies so that we could investigate and compare the performance of the three chosen algorithms on imbalanced conditions. Real-world H3 IAV data is inherently imbalanced, therefore we would consider the most robust models to be the ones that can handle imbalanced data.

Prior studies involving IAV classification have also used imbalanced datasets, although imbalanced to a much lesser extent in comparison to our study (Chrysostomou et al. 2021, Xu et al., 2021, Xu et al., 2022,).

Chrysostomou C, Alexandrou F, Nicolaou MA, Seker H. Classification of Influenza Hemagglutinin Protein Sequences using Convolutional Neural Networks. Proceedings of the Annual International Conference of the IEEE Engineering in Medicine and Biology Society, EMBS. 2021; 1682–1685. doi:10.1109/EMBC46164.2021.9630673

Xu Y, Wojtczak D. Predicting influenza a viral host using PSSM and word embeddings. 2021 IEEE Conference on Computational Intelligence in Bioinformatics and Computational Biology, CIBCB 2021. 2021. doi:10.1109/CIBCB49929.2021.9562959

Xu Y, Wojtczak D. Dive into machine learning algorithms for influenza virus host prediction with hemagglutinin sequences. Biosystems. 2022;220. doi:10.1016/J.BIOSYSTEMS.2022.104740

• Given that some classes have <30 samples (e.g., goose), any claim about high sensitivity should be treated with caution. Consider using metrics robust to imbalance such as macro-F1 and AUC per class.

We agree that class sensitivities should be interpreted with caution with an imbalanced dataset where some classes have a very low number of samples present. We now include macro-F1 in Table 3 and provide the F1 scores in Table S4.

3. Statistical rigor

• The manuscript reports high kappa and accuracy but does not provide sufficient validation to prove generalizability. Did you run independent external validation (e.g., on isolates from a different region or time period)?

Strong overall accuracy and kappa on the test set itself should be indicative of the generalizability of the models as the test set is unseen data, although the previously discussed issue with data leakage may lead to some questions regarding whether the performance metrics of high accuracy and kappa are overly optimistic.

To further validate our models for more generalizability, we assessed the model’s ability to detect between-species transmission through independent external validation of the first two case studies, which contained sequences that were not present in both the training and test datasets.

The first case study of canine H3N8 contains sequences only from the USA and no other canine H3N8 sequences are present in the training or test datasets. In this case study, the models effectively had 0% accuracy as they were misclassifying all segments of canine H3N8 as equine with high predicted probabilities. The models have not seen these canine H3N8s at all during training, resulting in them misclassifying them as equine as they were initially equine strains that were transmitted into canines.

The second case study of the H3N2 2010.2 clade in swine contains sequences only from the USA (although it has been detected throughout North America) and these sequences are not present in the training or test datasets either. A similar result was found in the second case study where the HA and NA segments also effectively had 0% accuracy for these segments as they were misclassifying the HA and NA as human and this 2010.2 clade had been identified to be a reassortant strain comprised of a human HA and NA.

Despite the models misclassifying both of these case studies, the 0% accuracy on these sequences can be flagged as sequences of interest, which could allow for identification of variants where between-species transmission has occurred if this pattern is found to be reoccurring.

Furthermore, the third environmental case study was another independent external validation for model generalization as these sequences do not have a known host, and only the accompanying literature could validate the model predictions. For this case study, there were around 6-7 instances of environmental sequences being predicted as swine with high predicted probabilities and the literature supported these predictions as being from agricultural swine fairs. A majority of the remaining was predicted as mallard with high predicted probability, but no accompanying literature was present to validate the model predictions.

• No confidence intervals or variance estimates for feature importance or predicted probabilities are provided.

Thank you for this insightful comment. We agree with the reviewer that these estimates would support the feature importance and predicted probabilities for our models. We initially did not think about possible ways to obtain these estimates as there are no base functions in the packages used that readily provide them. To obtain confidence intervals for the predicted probabilities, we performed stratified bootstrapping, although only with 50 replications and show the results of this with 2 randomly selected sequences from each class for the HA XGBoost model in the following table below as an example. The confidence intervals suggest high confidence in prediction of some sequences when the mean probability is very high or low under a range of different bootstrapped training datasets and much higher uncertainty as the mean probability is shifting to the mid-range, confirming further sets of sequences that would be of interest for this reason. We will look to incorporate these estimates for both feature importance and predicted probabilities within our future studies with a much larger number of replications, and will consider ways of effectively incorporating them into scientific communication.

SX Table. Mean predicted probabilities and 95% confidence intervals. Two sequences from each class in the test dataset were randomly selected and mean predicted probabilities and 95% confidence intervals for each class were calculated using 50 bootstrap replications for the HA XGBoost model.

Accession True Label Canine Chicken Equine Goose Human Mallard Swine

ON877627 Canine 0.995 [0.994–0.996] 0.001 [0.001–0.001] 0.001 [0.001–0.001] 0.001 [0.000–0.001] 0.001 [0.001–0.001] 0.001 [0.001–0.002] 0.001 [0.001–0.001]

KR154321 Canine 0.994 [0.991–0.995] 0.001 [0.001–0.001] 0.001 [0.001–0.001] 0.001 [0.000–0.001] 0.001 [0.001–0.001] 0.002 [0.001–0.002] 0.001 [0.001–0.003]

OQ292101 Chicken 0.000 [0.000–0.000] 0.998 [0.997–0.999] 0.000 [0.000–0.000] 0.000 [0.000–0.001] 0.001 [0.000–0.001] 0.000 [0.000–0.001] 0.000 [0.000–0.001]

OQ292789 Chicken 0.000 [0.000–0.000] 0.998 [0.996–0.998] 0.000 [0.000–0.000] 0.000 [0.000–0.000] 0.001 [0.000–0.001] 0.000 [0.000–0.001] 0.000 [0.000–0.001]

MH135223 Equine 0.001 [0.001–0.001] 0.001 [0.001–0.001] 0.991 [0.988–0.993] 0.001 [0.000–0.001] 0.001 [0.001–0.001] 0.003 [0.001–0.005] 0.004 [0.003–0.006]

OQ379779 Equine 0.001 [0.001–0.001] 0.001 [0.000–0.001] 0.991 [0.988–0.993] 0.001 [0.000–0.001] 0.001 [0.001–0.001] 0.001 [0.001–0.002] 0.005 [0.003–0.007]

CY138153 Goose 0.000 [0.000–0.001] 0.000 [0.000–0.001] 0.000 [0.000–0.001] 0.001 [0.000–0.001] 0.000 [0.000–0.001] 0.997 [0.995–0.999] 0.001 [0.000–0.002]

MT375535 Goose 0.004 [0.001–0.011] 0.010 [0.001–0.027] 0.004 [0.001–0.010] 0.388 [0.003–0.855] 0.018 [0.002–0.066] 0.571 [0.124–0.985] 0.005 [0.001–0.015]

KX415112 Human 0.000 [0.000–0.000] 0.000 [0.000–0.000] 0.000 [0.000–0.000] 0.000 [0.000–0.000] 1.000 [1.000–1.000] 0.000 [0.000–0.000] 0.000 [0.000–0.000]

OQ367467 Human 0.000 [0.000–0.000] 0.000 [0.000–0.000] 0.000 [0.000–0.000] 0.000 [0.000–0.000] 1.000 [1.000–1.000] 0.000 [0.000–0.000] 0.000 [0.000–0.000]

CY196065 Mallard 0.001 [0.000–0.003] 0.002 [0.000–0.006] 0.001 [0.000–0.004] 0.115 [0.000–0.456] 0.002 [0.000–0.006] 0.878 [0.532–0.999] 0.001 [0.000–0.004]

CY204177 Mallard 0.000 [0.000–0.000] 0.000 [0.000–0.000] 0.000 [0.000–0.000] 0.000 [0.000–0.001] 0.000 [0.000–0.000] 0.998 [0.998–0.999] 0.000 [0.000–0.000]

KC310691 Swine 0.000 [0.000–0.001] 0.000 [0.000–0.001] 0.000 [0.000–0.001] 0.000 [0.000–0.000] 0.003 [0.000–0.012] 0.000 [0.000–0.000] 0.996 [0.986–1.000]

KY284536 Swine 0.000 [0.000–0.001] 0.001 [0.000–0.002] 0.000 [0.000–0.001] 0.000 [0.000–0.001] 0.003 [0.001–0.009] 0.000 [0.000–0.001] 0.994 [0.986–0.998]

• The phylogenetic validation is interesting but is more descriptive than statistical; quantitative phylogenetic distance metrics would strengthen this.

We now state that we obtained patristic distances from the maximum likelihood tree on L225-L226 and state on L394-L397 that we provide summary statistics of mean, median, minimum, and maximum patristic distances in S8 Table, species-species patristic distances in S9 Table, and a figure showing the between and within class patristic distances for each of the 7 classes in S8 Fig.

4. Feature selection and reproducibility

• The feature space is huge (~22k features for some segments). Reducing to the top 10% of features ba

---

## [Decision Letter · Decision Letter 1]

22 Oct 2025

Evaluating machine learning approaches for host prediction using H3 influenza genomic data

PONE-D-25-31809R1

Dear Dr. Tran,

We’re pleased to inform you that your manuscript has been judged scientifically suitable for publication and will be formally accepted for publication once it meets all outstanding technical requirements.

Kind regards,

Victor C Huber

Academic Editor

PLOS ONE

Additional Editor Comments (optional):

Reviewers' comments:

Reviewer's Responses to Questions

**Comments to the Author**

Reviewer #2: All comments have been addressed

2. Is the manuscript technically sound, and do the data support the conclusions?

Reviewer #2: Yes

3. Has the statistical analysis been performed appropriately and rigorously?

Reviewer #2: Yes

4. Have the authors made all data underlying the findings in their manuscript fully available?

Reviewer #2: Yes

5. Is the manuscript presented in an intelligible fashion and written in standard English?

Reviewer #2: Yes

Reviewer #2: The authors have done an excellent job addressing the reviewers’ comments. The revised version and responses to my comments include a wealth of new data and thorough analyses, all clearly presented and well-discussed. The rationale behind the study and interpretation of results are sound and coherent. Overall, the manuscript is now strong and complete. I have no further questions or concerns.

**Do you want your identity to be public for this peer review?** For information about this choice, including consent withdrawal, please see our Privacy Policy

Reviewer #2: No

---

## [Editor Report · Acceptance letter]

PONE-D-25-31809R1

PLOS ONE

Dear Dr. Tran,

I'm pleased to inform you that your manuscript has been deemed suitable for publication in PLOS ONE. Congratulations! Your manuscript is now being handed over to our production team.

Kind regards,

on behalf of

Dr. Victor C Huber

Academic Editor

PLOS ONE